# Defective mesenchymal Bmpr1a-mediated BMP signaling causes congenital pulmonary cysts

**Yongfeng Luo[1]\*, Ke Cao[2], Joanne Chiu[1], Hui Chen[2], Hong-Jun Wang[2], Matthew E Thornton[3], Brendan H Grubbs[3], Martin Kolb[4], Michael S Parmacek[5], Yuji Mishina[6], Wei Shi[2]\*†**

[1]Department of Surgery, Children's Hospital Los Angeles, Keck School of Medicine, University of Southern California, Los Angeles, United States; [2]Division of Pulmonary, Critical Care & Sleep Medicine, Department of Internal Medicine, University of Cincinnati College of Medicine, Cincinnati, United States; [3]Division of Maternal Fetal Medicine, Department of Obstetrics and Gynecology, Keck School of Medicine, University of Southern California, Los Angeles, United States; [4]Department of Medicine, McMaster University, Hamilton, Canada; [5]Department of Medicine, Perelman School of Medicine, University of Pennsylvania, Philadelphia, United States; [6]Department of Biologic and Material Sciences, University of Michigan-Ann Arbor, Ann Arbor, United States

**\*For correspondence:**
yluo@chla.usc.edu (YL);
shiwe@ucmail.uc.edu (WS)

†Lead contact

**Abstract** Abnormal lung development can cause congenital pulmonary cysts, the mechanisms of which remain largely unknown. Although the cystic lesions are believed to result directly from disrupted airway epithelial cell growth, the extent to which developmental defects in lung mesenchymal cells contribute to abnormal airway epithelial cell growth and subsequent cystic lesions has not been thoroughly examined. In the present study using genetic mouse models, we dissected the roles of bone morphogenetic protein (BMP) receptor 1a (Bmpr1a)-mediated BMP signaling in lung mesenchyme during prenatal lung development and discovered that abrogation of mesenchymal *Bmpr1a* disrupted normal lung branching morphogenesis, leading to the formation of prenatal pulmonary cystic lesions. Severe deficiency of airway smooth muscle cells and subepithelial elastin fibers were found in the cystic airways of the mesenchymal *Bmpr1a* knockout lungs. In addition, ectopic mesenchymal expression of BMP ligands and airway epithelial perturbation of the Sox2-Sox9 proximal-distal axis were detected in the mesenchymal *Bmpr1a* knockout lungs. However, deletion of Smad1/5, two major BMP signaling downstream effectors, from the lung mesenchyme did not phenocopy the cystic abnormalities observed in the mesenchymal *Bmpr1a* knockout lungs, suggesting that a Smad-independent mechanism contributes to prenatal pulmonary cystic lesions. These findings reveal for the first time the role of mesenchymal BMP signaling in lung development and a potential pathogenic mechanism underlying congenital pulmonary cysts.

## eLife assessment

This **valuable** paper characterizes a murine model for congenital cystic airway abnormalities (CPAM). In contrast to previous assumptions that only epithelial cells are involved in the formation of pulmonary cysts, the authors provide **compelling** new evidence that defective BMP signaling in lung mesenchymal cells can disrupt airway development. Knowing that proper BMP signaling in mesenchymal cells is required for normal cyst-free lungs could potentially pave the way to understanding and preventing CPAM in infants at risk for this common disorder, which can be fatal if untreated. The

relevance of the murine model could be enhanced by further molecular and histological comparison with human cysts.

## Introduction

Congenital pulmonary cysts, resulting from abnormal fetal lung development, cause respiratory distress, infection, and pneumothorax in neonates. The dynamic pathogenic process and mechanisms are difficult to study in humans and few animal models are available. It is known that lung development begins with the specification of the respiratory domain in the ventral wall of the anterior foregut endoderm, as indicated by the expression of Nkx2-1, and the separation of the respiratory tract from the dorsal esophagus. The primary lung epithelial buds then undergo reiterated elongation and division to form the conducting airways (branching morphogenesis), followed by distal saccular formation and peripheral alveolarization to give rise to millions of gas exchange units (*McCulley et al., 2015*; *Morrisey and Hogan, 2010*). This well-known process literally describes the growth of respiratory epithelium based on the extensive studies done in the past decades. It is notable that lung morphogenesis is a complex process relying on the highly coordinated development of both lung epithelium and lung mesenchyme. Evidence has increasingly suggested that lung mesenchymal lineages, including airway and vascular smooth muscle cells (SMCs), pericytes, and stromal fibroblasts, are as important as epithelial cells in proper lung development (*Noe et al., 2019*; *Ren et al., 2016*; *Luo et al., 2015*). In addition to acting as a mechanical framework that supports the formation of the bronchi, bronchioles, and the distal alveoli, the lung mesenchyme provides a microenvironment that regulates the growth of epithelium through a variety of morphogenic signals, such as Wnts, fibroblast growth factors (Fgfs), and bone morphogenetic proteins (BMPs) (*McCulley et al., 2015*; *Nikolić et al., 2018*; *Morrisey et al., 2013*; *Luo et al., 2018*).

BMPs are a family of growth factors that direct many biological processes, including organogenesis (*Wang et al., 2014*). BMPs bind to the receptor complex of BMP receptor II (Bmpr2) and BMP receptor I (Bmpr1a or Bmpr1b), which in turn activates the intracellular Smad-dependent and Smad-independent pathways (*Shi and Massagué, 2003*). Mice with conventional *Bmpr1a* gene deletions are early embryonic lethal (E7.5–9.5) before lung organogenesis (*Mishina et al., 1995*). As reported by us and other groups, both hyperactivation and inhibition of BMP signaling in fetal lung epithelial cells result in lung malformation, which is primarily due to the defects in distal lung epithelial cell proliferation and differentiation/maturation (*Luo et al., 2016*; *Sun et al., 2008*; *Weaver et al., 1999*; *Bellusci et al., 1996*). However, the role of mesenchymal BMP signaling in regulating fetal lung development has not been studied due to the lack of lung mesenchyme-specific targeting tools in the past. We recently generated a lung mesenchyme-specific *Tbx4* lung enhancer-driven *loxP*/Cre mouse driver line (*Zhang et al., 2013*) which enables us to manipulate BMP signaling specifically in lung mesenchymal cells in vivo and to study its role in the lung development.

Although Bmpr1a is expressed predominantly in fetal mouse lung airway epithelial cells at the early gestation stage (embryonic day [E]12.5 to E14.5), its expression in fetal lung mesenchyme is also evident during mid-gestation (*Sun et al., 2008*). Herein, we specifically deleted *Bmpr1a* in fetal lung mesenchymal cells, which resulted in abnormal airway development and subsequent prenatal cystic malformation. This pathological phenotype resembles the features observed in pediatric patients diagnosed with congenital pulmonary airway malformation (CPAM). Therefore, investigating the abnormal lung phenotypes caused by lung mesenchyme-specific *Bmpr1a* knockout and revealing the underlying molecular and cellular mechanisms will significantly enhance our understanding of congenital lung diseases.

## Results

### Abrogation of Bmpr1a-mediated BMP signaling in lung mesenchyme disrupted fetal lung development

By crossing the *Tbx4-rtTA/TetO-Cre* driver line to the *floxed-Bmpr1a* mice (*Tbx4-rtTA/TetO-Cre/Bmpr1a*$^{fx/fx}$), *Bmpr1a* was specifically knocked out in the lung mesenchyme with doxycycline (Dox) induction from the beginning of lung formation (E6.5, *Figure 1—figure supplement 1A*). This was

**eLife digest** Congenital disorders are medical conditions that are present from birth. Although many congenital disorders are rare, they can have a severe impact on the quality of life of those affected.

For example, congenital pulmonary airway malformation (CPAM) is a rare congenital disorder that occurs in around 1 out of every 25,000 pregnancies. In CPAM, abnormal, fluid-filled sac-like pockets of tissue, known as cysts, form within the lungs of unborn babies. After birth, these cysts become air-filled and do not behave like normal lung tissue and stop a baby's lungs from working properly. In severe cases, babies with CPAM need surgery immediately after birth.

We still do not understand exactly what the underlying causes of CPAM might be. CPAM is not considered to be hereditary – that is, it does not appear to be passed down in families – nor is it obviously linked to any environmental factors. CPAM is also very difficult to study, because researchers cannot access tissue samples during the critical early stages of the disease.

To overcome these difficulties, Luo et al. wanted to find a way to study CPAM in the laboratory. First, they developed a non-human animal 'model' that naturally forms CPAM-like lung cysts, using genetically modified mice where the gene for the signaling molecule Bmpr1a had been deleted in lung cells.

Normally, Bmpr1a is part of a set of the molecular instructions, collectively termed BMP signaling, which guide healthy lung development early in life. However, mouse embryos lacking Bmpr1a developed abnormal lung cysts that were similar to those found in CPAM patients, suggesting that problems with BMP signalling might also trigger CPAM in humans.

Luo et al. also identified several other genes in the Bmpr1a-deficient mouse lungs that had abnormal patterns of activity. All these genes were known to be controlled by BMP signaling, and to play a role in the development and organisation of lung tissue. This suggests that when these genes are not controlled properly, they could drive formation of CPAM cysts when BMP signaling is compromised.

This work is a significant advance in the tools available to study CPAM. Luo et al.'s results also shed new light on the molecular mechanisms underpinning this rare disorder. In the future, Luo et al. hope this knowledge will help us develop better treatments for CPAM, or even help to prevent it altogether.

validated at the mRNA and protein levels (*Figure 1—figure supplement 1B and C*). For example, in E15.5 mesenchyme-specific *Bmpr1a* conditional knockout (CKO), Bmpr1a immunostaining was absent in lung mesenchyme but present in the epithelia (*Figure 1—figure supplement 1C*) while Bmpr1a was detected in both mesenchyme and epithelia of wildtype (WT) lungs. By gross view with quantitative analysis (*Figure 1A–C*), early embryonic lung morphogenesis between *Bmpr1a* CKO lungs and WT controls was comparable prior to E13.5. Decreased epithelial branching and increased size of branching tips were observed from E14.5. As lung development progressed, the terminal airways in *Bmpr1a* CKO lungs displayed dilation and exhibited increasingly extensive cystic changes. However, the overall size of the lungs remained relatively unchanged. The dynamic change of the lung cysts in *Bmpr1a* CKO lungs was further analyzed in hematoxylin and eosin (H&E)-stained tissue sections (*Figure 1D*). The walls of the lung cysts were lined with a single layer of epithelial cells, over the course of development, ultimately resulting in the formation of balloon-like cystic lesions by the end of gestation (E18.5). These findings were consistent with the overall morphology of the lungs. Interestingly, overall cell proliferation and apoptosis at E15.5, when lung cysts were developed, had no significant change between cystic *Bmpr1a* CKO and WT control lungs as measured by EdU labeling and TUNEL assay, respectively (*Figure 1E–G*).

## Deletion of Bmpr1a in fetal lung mesenchymal cells resulted in deficiency of airway-specific smooth muscle growth

To understand the molecular mechanisms underlying the aforementioned phenotypic changes, bulk RNA-seq was used to examine the differentially expressed genes (DEGs) between *Bmpr1a* CKO and WT lung tissues at E15.5. A total of 1001 DEGs (false discovery rate [FDR] $\leq$ 0.05 and $\log_2$FC [fold change] $\geq$1) were identified, among which 547 genes were upregulated and 454 genes were downregulated in *Bmpr1a* CKO lungs as compared to the transcriptome of WT lungs (*Figure 2A*). The raw

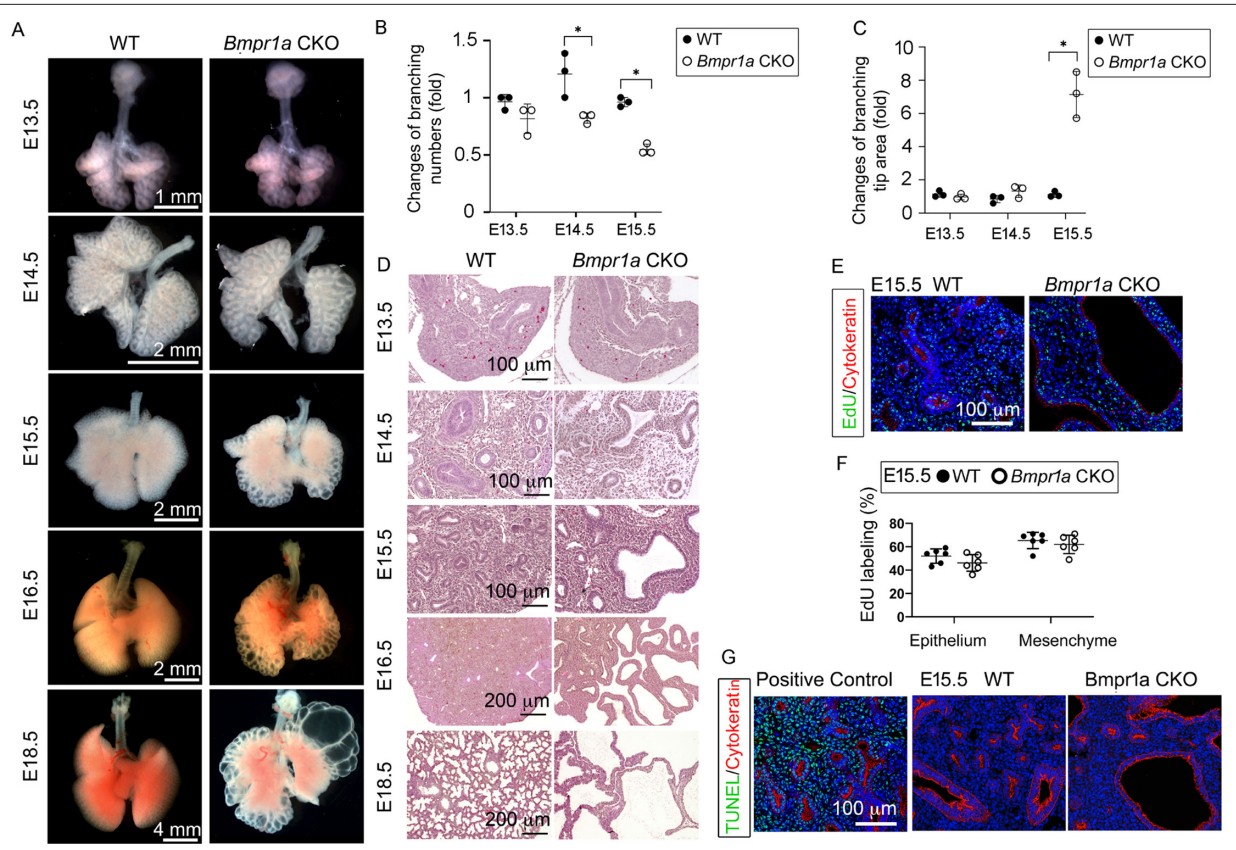

**Figure 1.** Lung mesenchyme-specific deletion of *Bmpr1a* caused abnormal lung morphogenesis and prenatal airway cystic lesions beginning in mid-gestation. (**A**) Brightfield images of whole wildtype (WT) and *Bmpr1a* conditional knockout (CKO) mouse lungs at different embryonic stages. (**B and C**) Quantitative measurement and comparison of terminal airway branching numbers and sizes. (**D**) Hematoxylin and eosin (H&E)-stained *Bmpr1a* CKO lungs at different embryonic stages. (**E and F**) EdU incorporation study for cell proliferation analysis in lung mesenchymal and cytokeratin-positive epithelial cells (n=6). (**G**) Apoptosis analysis by TUNEL assay. The positive control slides for apoptosis were generated by treating the tissue sections with DNase I. Pictures are representative of at least five samples in each condition.

The online version of this article includes the following source data and figure supplement(s) for figure 1:

**Figure supplement 1.** *Bmpr1a* was specifically deleted in lung mesenchymal cells.

**Figure supplement 1—source data 1.** Original file for the mRNA analysis in *Figure 1—figure supplement 1B* (exon 2 deletion of Bmpr1a).

**Figure supplement 1—source data 2.** PDF containing *Figure 1—figure supplement 1B* and original image of the relevant mRNA analysis (exon 2 deletion of Bmpr1a).

data are available in the Gene Expression Omnibus repository under accession number GSE97946. Gene Ontology (GO) enrichment analysis showed that lung mesenchymal *Bmpr1a* regulates a large number of genes involved in the Muscle System Process (p<0.001, *Figure 2B*).

As shown above in *Figure 1*, enlarged airways were initially seen in *Bmpr1a* CKO lungs from E14.5. Whole mount staining of smooth muscle actin and comprehensive 3D imaging revealed a specific absence of airway SMCs surrounding the dilated airway branching in *Bmpr1a* CKO lungs, while unaffected airway branches in *Bmpr1a* CKO lungs remained surrounded by SMCs, as compared to the WT lungs (*Figure 2C*). As lung development proceeded to E15.5, the distal airway enlargement became increasingly severe and extensive, as indicated by the whole mount staining of Cdh1 (E-cadherin), which outlines the airway epithelia. Reduction and even lack of airway SMCs was consistently found around the enlarged airways of E15.5 *Bmpr1a* CKO lungs, shown by immunostaining for SMC markers Acta2 and Myh11 (*Figure 2D and E*). In contrast, vascular SMCs in *Bmpr1a* CKO lungs appeared to remain unaffected (*Figure 2E*).

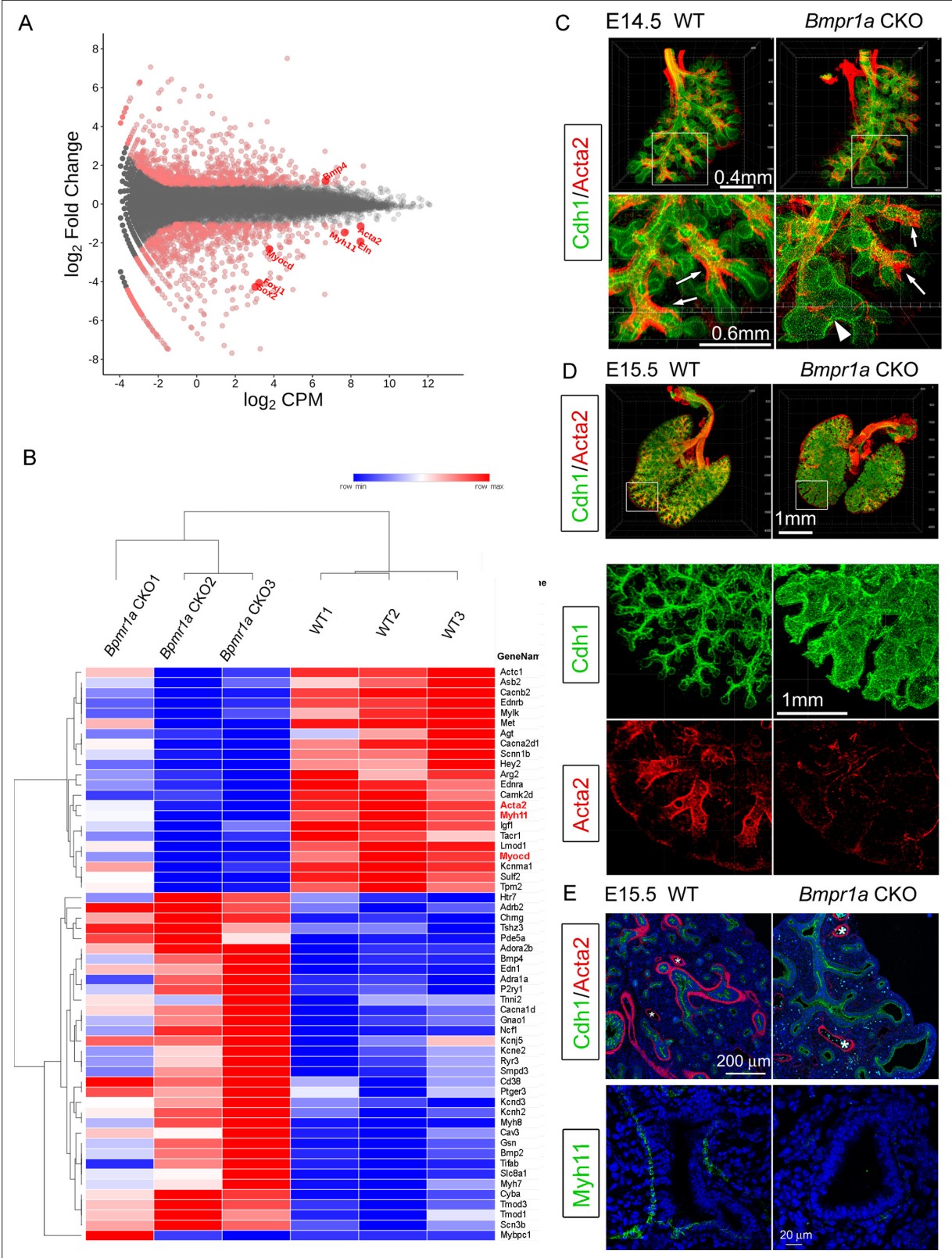

**Figure 2.** Airway smooth muscle was substantially reduced in *Bmpr1a* conditional knockout (CKO) lungs. (**A**) Scatter plots of RNA-seq analysis for differentially expressed genes (DEGs) (n=3). (**B**) RPKM heatmap of Muscle System Process Genes (GO:0003012). Genes involved in the Muscle System Process with significant changes between *Bmpr1a* CKO and wildtype (WT) lungs are shown in the RNA-seq heatmap. The input data was RPKM values generated from the raw data using the R/Bioconductor program 'edgeR'. (**C and D**) Whole mount immunostaining of Cdh1 and Acta2 in embryonic day

*Figure 2 continued on next page*

*Figure 2 continued*

(E)14.5 and E15.5 lungs. Excessive dilation of airway terminals and extensive reduction of airway smooth muscle were observed in E15.5 *Bmpr1a* CKO lungs. The normal-looking airways are marked by arrows and the enlarged epithelial bud accompanied with compromised smooth muscle are marked by arrowheads. (**E**) E15.5 lung section stained with Cdh1& Acta2 or Myh11. *Vascular structures.

## The effect of Bmpr1a deletion on the other components of lung mesenchyme

The *Tbx4* driver line also targets lung pericytes and endothelial cells with early induction (*Zhang et al., 2013*). However, *Bmpr1a* deletion in these cell lineages had no effects on these cells or the associated pulmonary vasculature, as measured by the relevant cellular markers (Cspg4 and Pecam1) at both the mRNA and protein levels (*Figure 3A and B*). Emerging evidence has suggested that the extracellular matrix (ECM), including the airway basement membrane and matrix fibril collagen, is critical to lung development and maturation (*Luo et al., 2018*; *Jiang et al., 2019*). Several major components of the ECM in the *Bmpr1a* CKO lungs, such as laminin and type III collagen, were examined at both the mRNA and protein levels. No significant changes were found, even in the areas surrounding the dilated airways and lung cysts when compared to the WT lungs (*Figure 3A and B*).

However, the mesenchymal knockout of *Bmpr1a* caused significant reduction of elastin expression, as detected by real-time PCR and western blot (*Figure 3A and D*). In the *Bmpr1a* CKO lungs, co-immunostaining of epithelia (Cdh1), SMCs (Acta2), and elastin (Eln) showed that this decrease in elastin expression occurred primarily in the areas adjacent to the epithelia of enlarged airways with the deficiency of airway smooth muscle. This suggests that the elastin abnormality may be caused by airway SMC deficiency or by reduced elastin expression in adjacent epithelial cells (*Figure 3C*). To determine whether BMP signaling can directly regulate elastin expression in fetal lung mesenchymal cells, alteration of elastin protein expression upon BMP4 treatment in primary fetal lung mesenchymal cells was analyzed. A significant increase in elastin protein was observed in the cells treated with BMP4 (*Figure 3E and F*). However, inclusion of the BMP type 1 receptor (BMPR1) inhibitor LDN193189 (LDN) fully blocked the BMP4-stimulated elastin upregulation.

## Bmpr1a-mediated BMP signaling regulated airway SMC differentiation through a Smad-independent pathway

Activation of the BMP receptor complex can trigger intracellular Smad-dependent and Smad-independent pathways (*Shi and Massagué, 2003*). In line with this report, the downregulation of Smad1/5-mediated BMP signaling was also observed in the *Bmpr1a* CKO lungs, evident from the reduced phosphorylation of Smad1/5 (p-Samd1/5 in *Figure 4A*). The role of Bmpr1a in mediating Smad1/5 activation in lung mesenchymal cells was further investigated in primary culture of fetal lung mesenchymal cells isolated from WT fetal lung tissues. Treatment of these cells with BMP4 activated Smad1/5 phosphorylation, while the addition of BMPR1 inhibitor LDN reduced the BMP4-induced Smad1/5 phosphorylation (*Figure 4E*). To determine whether the Smad1/5-mediated BMP canonical pathway controls airway SMC growth in vivo, *Smad1* and *Smad5* were both deleted in lung mesenchyme by crossing the floxed-*Smad1* and -*Smad5* mice with the *Tbx4-rtTA/TetO-Cre* driver. Although abnormal fetal lungs with hypoplastic development were present in the *Smad1/5* double CKO mice, airway dilation and cystic lesions were not detected (*Figure 4B and C*). Furthermore, in the *Smad1/5* CKO lungs, airway SMC growth was not affected, as indicated by the expression of Acta2 around the airways (*Figure 4D*). This different phenotypic observation suggests that airway SMC growth may be independent of Smad1/5-mediated pathway.

We then examined other major Smad-independent pathways in the *Bmpr1a* CKO lungs. Reduced phosphorylation of p38 (p-p38) mitogen-activated protein kinase (MAPK), but not Jnk and Erk, was detected in the *Bmpr1a* CKO lungs (*Figure 4A*). Bmp4 treatment of the cultured primary lung mesenchymal progenitor cells also specifically activated p38, a response that was specifically blocked by the addition of the BMPR1 inhibitor LDN (*Figure 4E*). Additionally, the role of the BMP4-Bmpr1a-p38 pathway in regulating SMC-related gene expression was assessed in cultured fetal lung mesenchymal cells. As previously reported (*Young et al., 2020*), Myocd is a key transcription co-factor in controlling airway SMC differentiation. Myh11 and Acta2, major contractile proteins, are two prominent SMC markers. Treatment of lung mesenchymal progenitor cells with BMP4 induced a pronounced increase

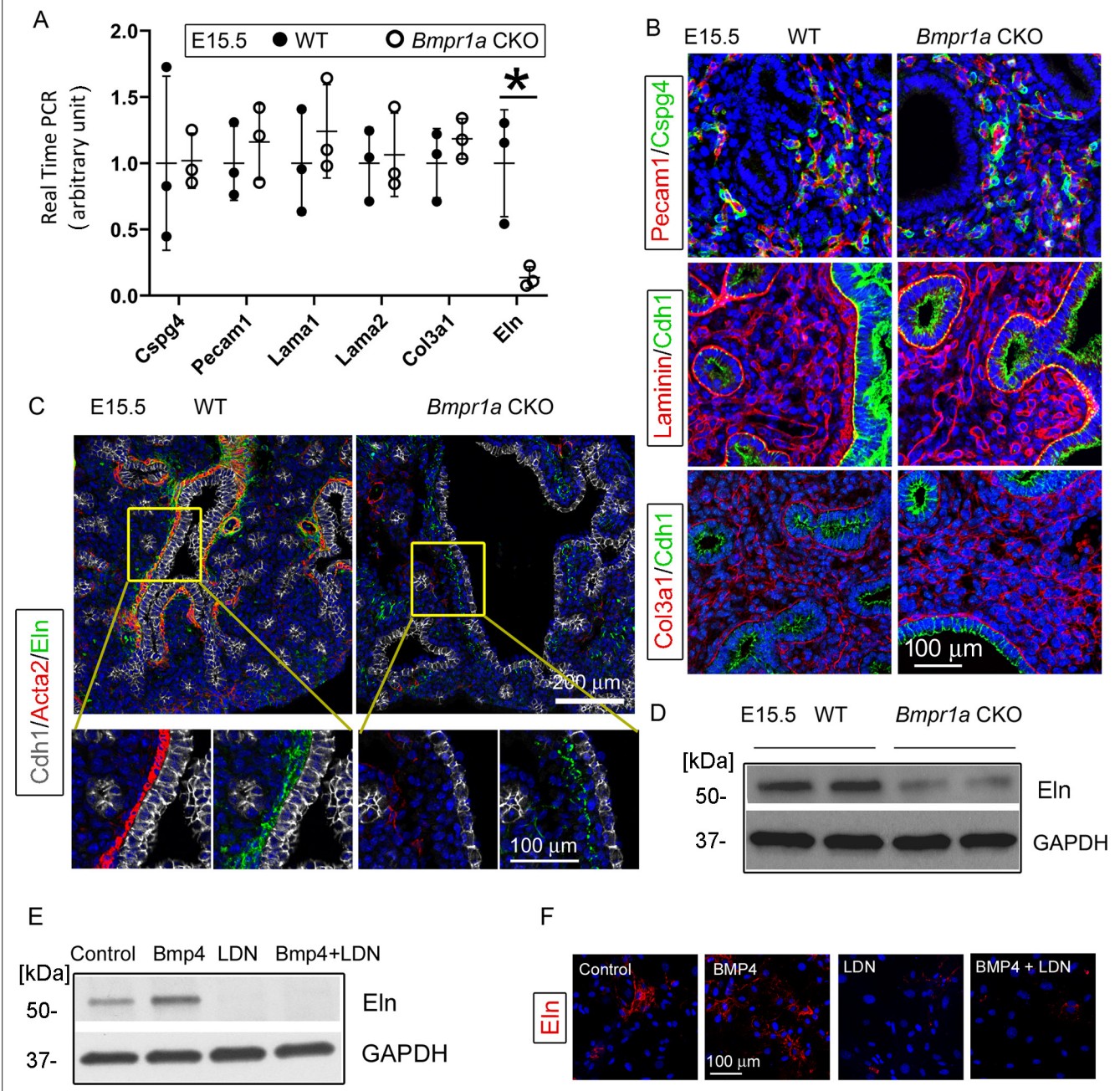

**Figure 3.** *Bmpr1a* conditional knockout (CKO) resulted in a significant reduction in elastin expression underneath the airway epithelia of embryonic day (E)15.5 lungs, but not in the pericytes, vasculature, and basement membrane. (**A**) Expression of *Cspg4*, *Pecam1*, *Lama1*, *Lama2*, *Col3a1,* and *Eln* at the mRNA level was measured by real-time PCR, *p<0.05. (**B**) E15.5 lung section stained with Cspg4, Pecam1, laminin, and Col3a1. (**C**) E15.5 lung section stained with Cdh1, Acta2, and elastin. (**D**) Reduced elastin expression of *Bmpr1a* CKO lungs at the protein level was detected by western blot (WB). (**E and F**) Elastin expression in E15.5 lung mesenchymal cells was upregulated by BMP4 and downregulated by BMP type 1 receptor-specific inhibitor LDN193189 (LDN), as detected by WB and immunofluorescence staining.

The online version of this article includes the following source data for figure 3:

**Source data 1.** Original file for the western blot (WB) analysis in *Figure 3D* (anti-Eln and anti-GAPDH).

**Source data 2.** PDF containing *Figure 3D* and original scans of the relevant western blot (WB) analysis (anti-Eln and anti-GAPDH) with highlighted bands and sample labels.

**Source data 3.** Original file for the western blot (WB) analysis in *Figure 3E* (anti-Eln).

**Source data 4.** Original file for the western blot (WB) analysis in *Figure 3E* (anti-GAPDH).

*Figure 3 continued on next page*

*Figure 3 continued*

**Source data 5.** PDF containing *Figure 3E* and original scans of the relevant western blot (WB) analysis (anti-Eln and anti-GAPDH) with highlighted bands and sample labels.

in the expression of these SMC-related genes at both the mRNA and protein levels (*Figure 4F and G*). Inhibition of either Bmpr1a by LDN or p38 by SB203580 (SB) significantly attenuated the promoting effect of BMP4 on SMC differentiation.

## Bmpr1a-mediated Bmp signaling specifically promoted non-vascular SMC growth

As shown in *Figure 2E*, *Bmpr1a* CKO caused defective growth of airway SMCs in vivo, while vascular SMCs remained unaffected. To investigate the distinct effects of the Bmpr1a-mediated pathway on non-vascular SMCs vs. vascular SMCs, primary fetal lung mesenchymal cells were isolated from *Tagln-YFP/Cspg4-DsRed* double reporter mice, in which airway SMCs/myofibroblasts were solely labeled with YFP expression while vascular SMCs and pericytes were marked by both YFP and DsRed expression (*Figure 5A*). Following fluorescence-activated cell sorting (FACS) of the single-cell suspension obtained from dissociated E15.5 lung tissues (*Figure 5B*), the vascular SMCs (YFP$^+$/DsRed$^+$) were separated from non-vascular SMCs (YFP$^+$/DsRed$^-$). These two distinct SMC populations were then cultured for further analysis (*Paez-Cortez et al., 2013*). As shown in *Figure 5C*, BMP4 treatment could promote SMC-associated gene expression in lung SMCs of non-vascular origin rather than those of vascular origin, as assessed by their Myh11 expression. This BMP4-induced upregulation of Myh11 expression in non-vascular SMCs was effectively blocked by concurrent treatment with the BMP type 1 receptor-specific inhibitor LDN.

## Loss of airway SMCs alone was not sufficient to cause prenatal cystic malformation in vivo

Previous studies using embryonic lung explant culture have suggested that airway SMCs play an important role in branching morphogenesis in developmental lungs (*Kim et al., 2015*; *Jesudason, 2009*). *Myocd* encodes an important transcriptional coactivator of serum response factor, modulating the expression of smooth muscle-specific cytoskeletal and contractile proteins (*Parmacek, 2007*). In our *Bmpr1a* CKO lungs, expression of *Myocd* was substantially downregulated (*Figure 6A*). To determine whether Bmpr1a-mediated downregulation of Myocd expression and the resultant airway SMC defects directly caused fetal airway cystic lesions in vivo, *Myocd* was deleted in lung mesenchyme starting at E6.5 using the same *Tbx4-rtTA/TetO-Cre* driver (*Figure 6—figure supplement 1*). As detected by Acta2 immunostaining, knockout of *Myocd* led to a severe deficiency in airway SMC development (*Figure 6B*) resembling the airway SMC deficiency observed in the *Bmpr1a* CKO lungs (*Figures 2 and 3C*). However, mesenchymal Bmpr1a expression remained unchanged in the *Myocd* CKO lungs (*Figure 6C*). No significant branching defect or cystic malformation was observed in the *Myocd* CKO lungs (*Figure 6D*), which is consistent with the previous report (*Young et al., 2020*). Despite severe disruption of airway SMC development in the *Myocd* CKO lungs, elastin fiber deposition in the airways was unaffected (*Figure 6B*), which was a significant difference when compared to the findings in the *Bmpr1a* CKO airways, where a deficiency of elastin fibers adjacent to the epithelium was observed (*Figure 3C*).

## The proximal-distal developmental patterning of fetal lung epithelia was perturbed in the cystic lungs of the *Bmpr1a* CKO mice

The coordinated airway epithelial differentiation that forms the proximal-distal axis is another important mechanism controlling lung morphogenesis (*Herriges and Morrisey, 2014*). The proximal epithelial cells, marked by the expression of several unique genes such as *Sox2*, serve as progenitors for neuroendocrine cells, ciliated epithelial cells, and Club cells in the bronchi/bronchioles, while the distal epithelial cells, defined by expression of *Sox9*, *Id2*, and *Sftpc* in fetal lungs, are the progenitors of peripheral alveolar epithelial cells (*Treutlein et al., 2014*). The mechanisms by which mesenchymal cells regulate airway epithelial cell fate during lung development are largely unknown. Interestingly, in the *Bmpr1a* CKO lung, Sox2-positive epithelial cells were barely detected in the proximal airways.

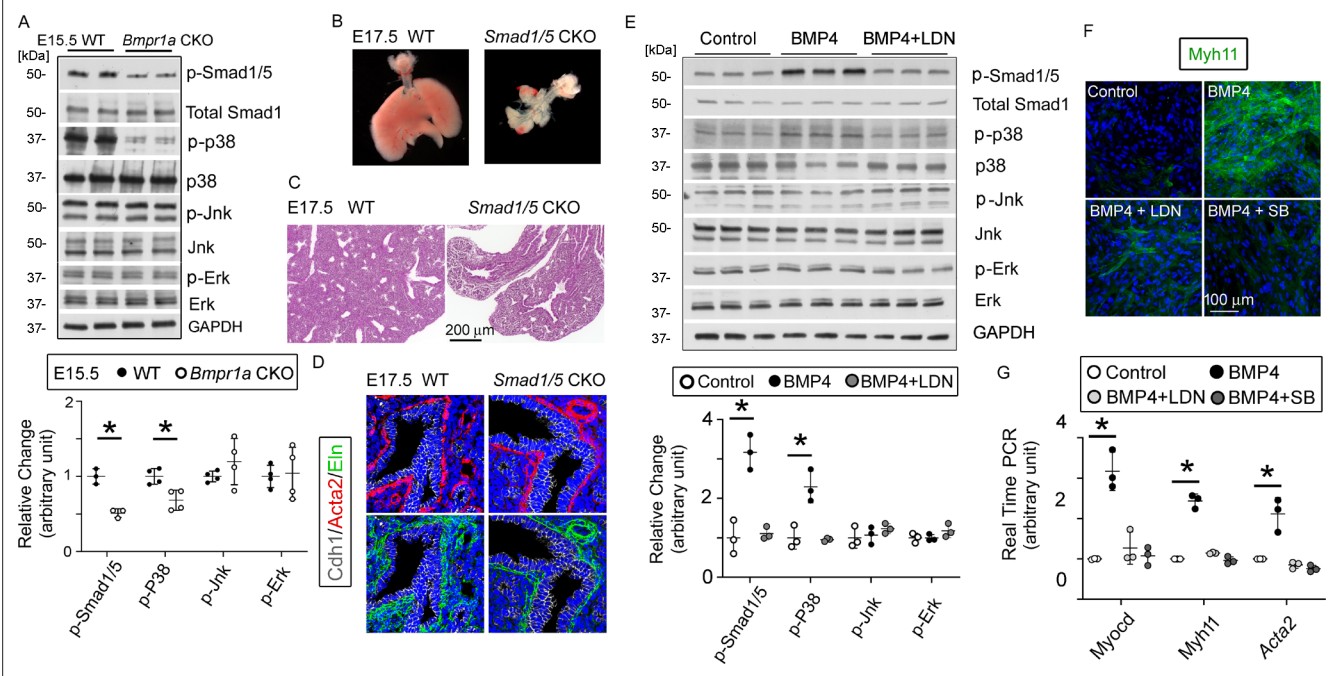

**Figure 4.** The bone morphogenetic protein (BMP) pathway regulates the myogenesis of lung mesenchymal cells via the Smad-independent pathway. (**A**) Activation of the intracellular downstream Smad1, p38, Jnk, and Erk signaling pathways in wildtype (WT) and *Bmpr1a* conditional knockout (CKO) lung tissues was detected by the western blot (WB) and quantified by densitometry. The levels of protein phosphorylation were normalized by the corresponding total protein and is presented as a relative change to the WT, *p<0.05. (**B**) Gross view of whole lungs from *Smad1/5* double conditional knockout mice (*Smad1/5* CKO) and WT littermates showed that simultaneous deletion of *Smad1* and *Smad5* in lung mesenchyme completely disrupted lung development. (**C**) No airway dilation or cysts were observed in the hematoxylin and eosin (H&E)-stained tissue sections of the *Smad1/5* CKO lungs at embryonic day (E)17.5. (**D**) Expression of airway smooth muscle cells (SMCs) and elastin was not altered in E17.5 *Smad1/5* CKO lungs, as shown by immunostaining of Cdh1, Acta2, and elastin. (**E**) Changes of intracellular signaling pathways in cultured fetal lung mesenchymal cells upon treatment with BMP4 (50 ng/ml) and/or LDN193189 (200 nM) were detected by WB and quantified by densitometry. The relative change to the control condition is presented, *p<0.05. (**F and G**) Altered expression of SMC genes at the protein level (Myh11) and the mRNA level (*Myocd, Myh11, and Acta2*) was respectively analyzed by immunostaining and real-time PCR for the primary culture of E15.5 WT lung mesenchymal cells treated with BMP4 (50 ng/ml), LDN (200 nM), and SB (1 μM), *p<0.05.

The online version of this article includes the following source data for figure 4:

**Source data 1.** Original file for the western blot (WB) analysis in *Figure 4A* (anti-p-Smad1/5).

**Source data 2.** Original file for the western blot (WB) analysis in *Figure 4A* (anti-total Smad1, anti-Jnk, and anti-Erk).

**Source data 3.** Original file for the western blot (WB) analysis in *Figure 4A* (anti-p-p38).

**Source data 4.** Original file for the western blot (WB) analysis in *Figure 4A* (anti-p38).

**Source data 5.** Original file for the western blot (WB) analysis in *Figure 4A* (anti-p-Jnk).

**Source data 6.** Original file for the western blot (WB) analysis in *Figure 4A* (anti-p-Erk).

**Source data 7.** Original file for the western blot (WB) analysis in *Figure 4A* (anti-GAPDH).

**Source data 8.** PDF containing *Figure 4A* and original scans of the relevant western blot (WB) analysis (anti-p-Smad1/5, anti-total Smad1/5, anti-p-p38, anti-p38, anti-p-Jnk, anti-Jnk, anti-p-Erk, anti-Erk, and anti-GAPDH) with highlighted bands and sample labels.

**Source data 9.** Original file for the western blot (WB) analysis in *Figure 4E* (anti-p-Smad1/5).

**Source data 10.** Original file for the western blot (WB) analysis in *Figure 4E* (anti-total Smad1).

**Source data 11.** Original file for the western blot (WB) analysis in *Figure 4E* (anti-p-p38).

**Source data 12.** Original file for the western blot (WB) analysis in *Figure 4E* (anti-p38).

**Source data 13.** Original file for the western blot (WB) analysis in *Figure 4E* (anti-p-Jnk).

**Source data 14.** Original file for the western blot (WB) analysis in *Figure 4E* (anti-Jnk).

**Source data 15.** Original file for the western blot (WB) analysis in *Figure 4E* (anti-p-Erk).

**Source data 16.** Original file for the western blot (WB) analysis in *Figure 4E* (anti-Erk).

*Figure 4 continued on next page*

*Figure 4 continued*

**Source data 17.** Original file for the western blot (WB) analysis in *Figure 4E* (anti-GAPDH).

**Source data 18.** PDF containing *Figure 4E* and original scans of the relevant western blot (WB) analysis (anti-p-Smad1/5, anti-total Smad1/5, anti-p-p38, anti-p38, anti-p-Jnk, anti-Jnk, anti-p-Erk, anti-Erk, and anti-GAPDH) with highlighted bands and sample labels.

Similarly, lack of further differentiation of Foxj1-positive cells was also observed. In contrast, cells expressing Sox9 and Sftpc, which are normally confined to the distal epithelial cells during lung branching, were expanded to the proximal region of the airways in *Bmpr1a* CKO lungs (*Figure 7A*). The analysis of tissue RNA-seq data revealed that the mesenchymal *Bmpr1a* knockout downregulated multiple proximal airway epithelial marker genes, while various distal airway epithelial marker genes were upregulated (*Figure 7B*). Ectopic expression of Bmp4 in fetal mouse lung has been reported to cause distal epithelial expansion (*Bellusci et al., 1996*). Increased expression of *Bmp4* in E15.5 *Bmpr1a* CKO lungs was initially found by RNA-seq (logFC>1, GEO#: GSE97946) and validated by RT-PCR (*Figure 7C*). The inverse relationship between Bmpr1a deletion (intracellular BMP signaling) and Bmp4 ligand expression in these fetal lung mesenchymal progenitor cells was further confirmed by comparing Bmp4 expression in isolated fetal lung mesenchymal cells with different *Bmpr1a* genotypes (*Figure 7D and E*). Alterations of other key pathways including *Fgf*, *Wnt*, and *Shh* were not detected by RNA-seq and RT-PCR through analyzing the related targeted genes.

## Discussion

Our current study provides solid in vivo evidence for the first time that Bmpr1a-mediated BMP signaling in lung mesenchymal cells is indispensable in lung development. Lung mesenchymal deletion of *Bmpr1a* at an early developmental stage causes abnormal branching morphogenesis and then severe lung cysts at late gestation in mice. Interestingly, congenital pulmonary cysts are reported in human CPAM patients as a result of abnormal prenatal lung development (*Stocker, 2009*). Reduction

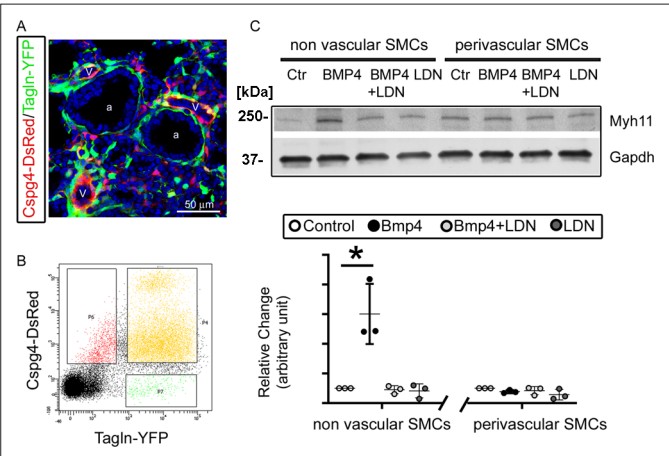

**Figure 5.** Bmpr1a-mediated signaling played different roles in the differentiation of airway versus vascular smooth muscle cells (SMCs). (**A**) YFP and DsRed expression patterns in the fetal lungs of *Tagln-YFP/Cspg4-DsRed* mice. a: airway, v: vessel. (**B**) Airway SMCs (YFP[+]) and vascular SMCs (YFP[+]/DsRed[+]) were isolated by fluorescence-activated cell sorting (FACS). (**C**) The differential effect of Bmp4 treatment (50 ng/ml) on contractile protein Myh11 expression was analyzed in SMCs of non-vascular origin (YFP[+]) versus vascular origin (YFP[+]/DsRed[+]), and the role of Bmpr1a in mediating this effect was tested by adding its specific inhibitor LDN193189 (200 nM). The western blot (WB) data of Myh11 expression was normalized to GAPDH (loading control) and is represented as a relative change to the control condition, *p<0.05.

The online version of this article includes the following source data for figure 5:

**Source data 1.** Original file for the western blot (WB) analysis in *Figure 5C* (anti-Myh11).

**Source data 2.** Original file for the western blot (WB) analysis in *Figure 5C* (anti-GAPDH).

**Source data 3.** PDF containing *Figure 5C* and original scans of the relevant western blot (WB) analysis (anti-Myh11 and anti-GAPDH) with highlighted bands and sample labels.

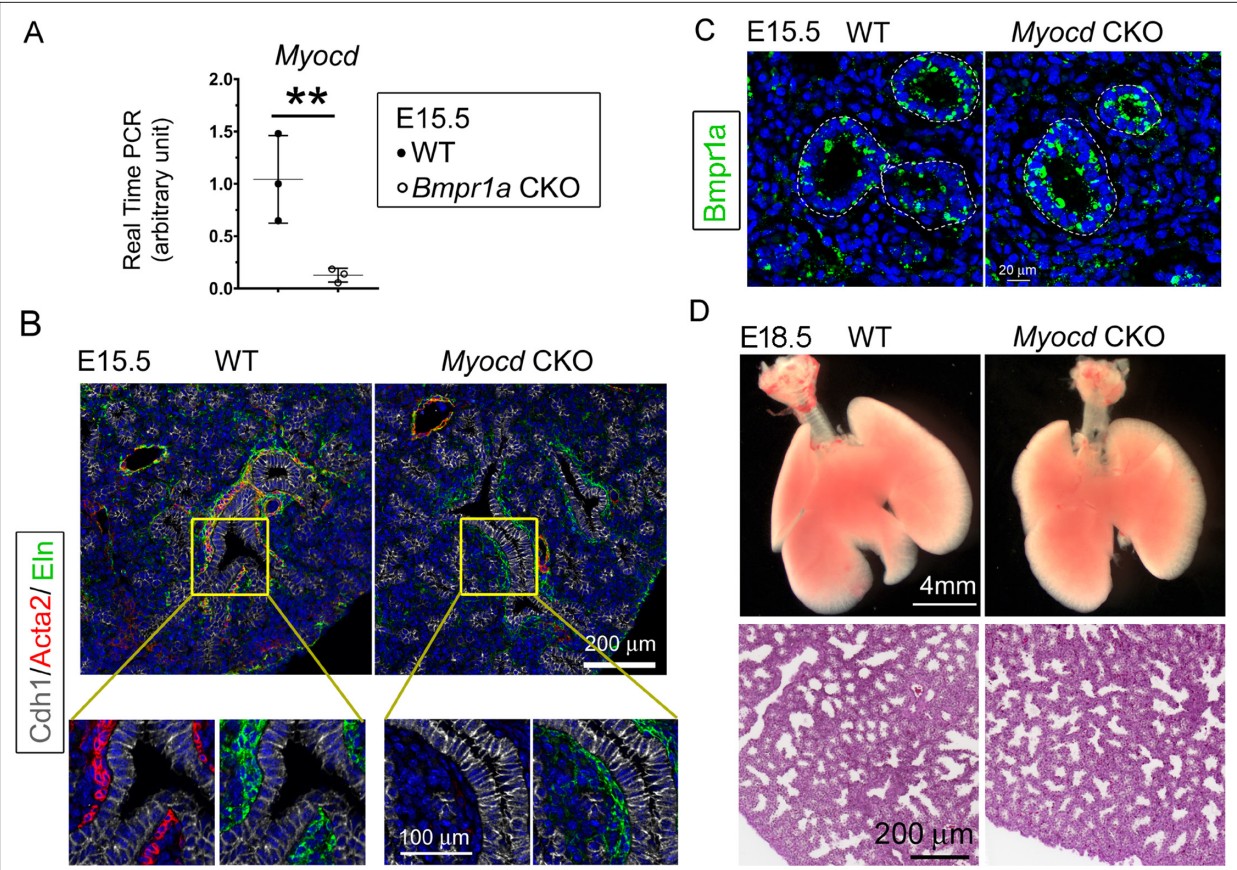

**Figure 6.** Lung mesenchymal knockout of *Myocd* did not cause any branching abnormalities or lung cysts. (**A**) The expression of *Myocd* in *Bmpr1a* conditional knockout (CKO) lungs was substantially decreased, as measured by real-time PCR, **p<0.01. (**B**) Deficiency in airway smooth muscle cells (SMCs) was observed in embryonic day (E)15.5 mesenchyme-specific *Myocd* CKO lungs, as shown by co-immunofluorescence staining of Cdh1, Acta2, and elastin. (**C**) Comparison of Bmpr1a expression between E15.5 *Myocd* CKO and wildtype (WT) control lungs by immunofluorescence staining. (**D**) Comparison of the lungs between WT and *Myocd* CKO mice at the end of gestation (E18.5) did not reveal any significant morphological changes by gross view. No histological difference was found between the WT and the *Myocd* CKO lungs by examining their hematoxylin and eosin (H&E)-stained lung tissue sections.

The online version of this article includes the following figure supplement(s) for figure 6:

**Figure supplement 1.** Schematic representation of the lung mesenchyme-specific knockout of *Myocd* by *Tbx4-rtTA/Teto-Cre* driver line.

of airway SMCs and decreased subepithelial elastin fibers are also found to be the common pathological changes in the cystic airways of human CPAM samples (*Jiang et al., 2019*). Although the congenital pulmonary cysts in CPAM patients can be identified by a prenatal ultrasound examination, the tissue specimens for pathological analysis are only available from postnatal surgery, when advanced airway cysts have already developed. This makes it difficult to study the pathogenic molecular mechanisms that trigger early cyst formation during fetal lung development in human subjects. Studies in mouse models suggest that altered growth factor signaling in a prenatal time window is sufficient to cause cyst formation (*Miao et al., 2021*). The abnormal BMP signaling pathway may be involved in the initiation of CPAM, but not in the later stage when cystic lesions have formed. Interestingly, integrated suppression of BMP signaling pathway was recently found in human CPAM specimens by an RNA-seq approach (*Zhang et al., 2022*). Therefore, the lung mesenchymal *Bmpr1a* knockout model developed here is a potentially useful tool for studying the pathogenic process of CPAM.

It has been reported that epithelial overexpression of Bmp4 appears to induce extensive mesenchymal SMC differentiation in vivo (*Badri et al., 2008*), and that BMP4 promotes myocyte differentiation and inhibits cell proliferation in cultured human fetal lung fibroblasts via the Smad1 pathway (*Jeffery et al., 2005*). Smad1 and Smad5 are critical downstream effectors of BMP type I receptors, mediating BMP canonical signal pathway (*Wang et al., 2014*; *Katagiri and Watabe, 2016*).

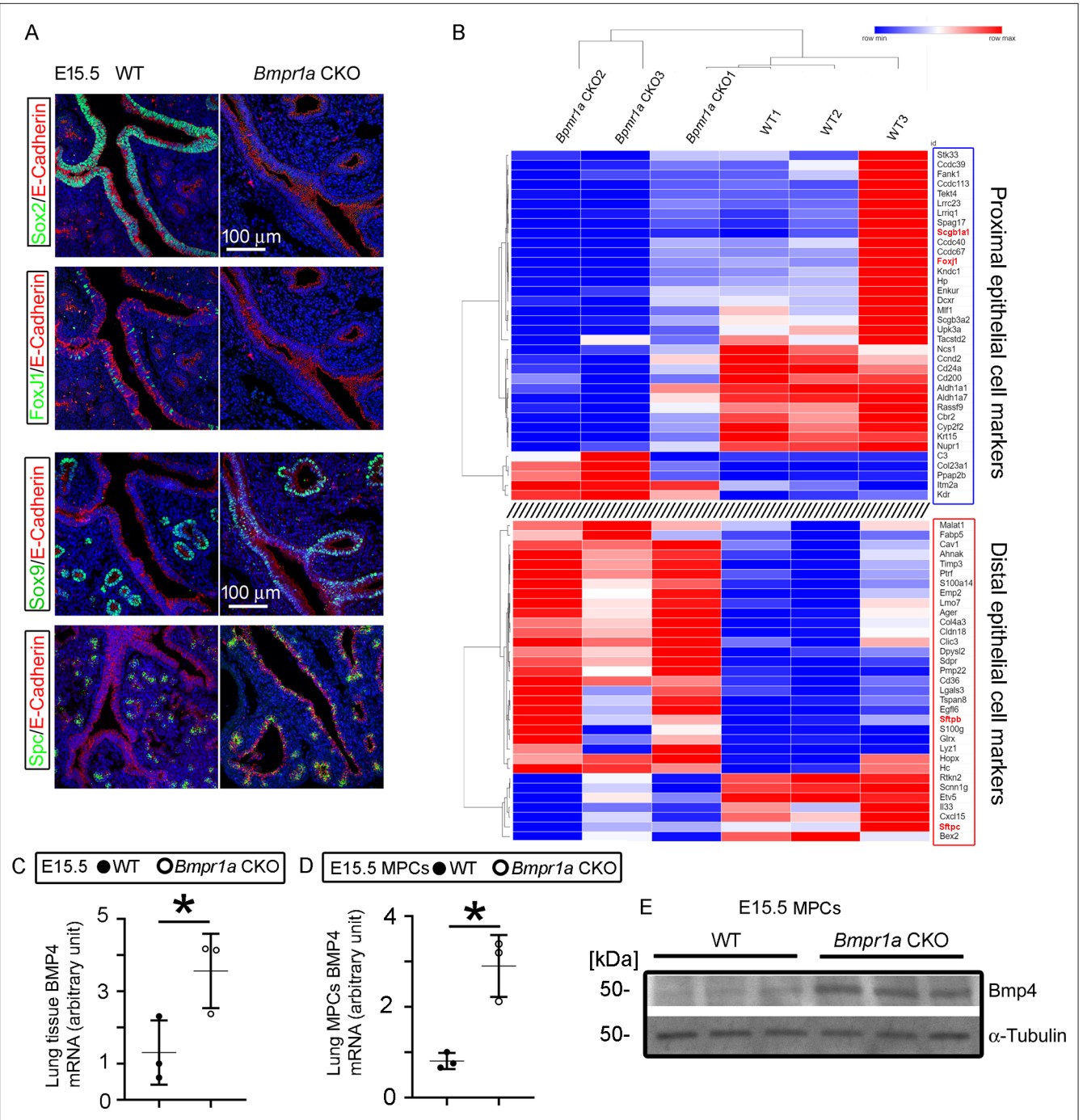

**Figure 7.** Mesenchymal *Bmpr1a* deletion disrupted airway epithelial proximal-distal differentiation and development. (**A**) Proximal epithelial cells, marked by Sox2 and Foxj1 staining, were significantly decreased in the proximal portion of the airways in embryonic day (E)15.5 *Bmpr1a* conditional knockout (CKO) lungs. Ectopic distribution of distal epithelial cells marked by Sox9 and Spc staining was detected in the proximal airways of E15.5 *Bmpr1a* CKO lungs. (**B**) Heatmap of RNA-seq data showing significant changes in the marker genes of proximal and distal epithelial cells. (**C**) Increased *Bmp4* expression at the mRNA level was detected in E15.5 *Bmpr1a* CKO lung tissue by RT-PCR. (**D and E**) Bmp4 expression in isolated fetal lung mesenchymal cells with genotypes of wildtype (WT) vs. *Bmpr1a* CKO was analyzed at both the mRNA and protein levels by RT-PCR and western blot (WB) respectively.

The online version of this article includes the following source data for figure 7:

**Source data 1.** Original file for the western blot (WB) analysis in *Figure 7E* (anti-Bmp4).

**Source data 2.** Original file for the western blot (WB) analysis in *Figure 7E* (anti-α-tubulin).

*Figure 7 continued on next page*

*Figure 7 continued*

**Source data 3.** PDF containing *Figure 7E* and original scans of the relevant western blot (WB) analysis (anti-Bmp4 and anti-α-tubulin) with highlighted bands and sample labels.

However, our in vivo study indicates that simultaneous deletion of *Smad1* and *Smad5* in mouse lung mesenchyme does not affect the differentiation of airway SMCs although it disrupts early embryonic lung morphogenesis. This suggests that the decrease in phosphorylated Smad1 and Smad5 in the *Bmpr1a* CKO lungs is not responsible for the defective airway SMC growth. In addition to the Smad-dependent BMP signal pathway, BMP-activated Bmp receptors also function through MAPK (*Wang et al., 2014*; *Katagiri and Watabe, 2016*; *von Bubnoff and Cho, 2001*). Many studies have shown that p38 MAPK regulates cell proliferation in a variety of SMC types, including airway SMCs (*Chen and Khalil, 2006*; *Pelaia et al., 2008*). The p38 MAPK is also involved in myogenic differentiation and myofibroblast trans-differentiation (*Choi et al., 2011*; *Meyer-Ter-Vehn et al., 2006*; *Sebe et al., 2008*). In the present study, reduced phosphorylation of p38 was detected in mesenchymal *Bmpr1a* CKO lungs while addition of BMP4 activated the p38 signal pathway in mouse primary fetal lung mesenchymal cells. Moreover, the myogenic response to BMP4 stimulation in the cultured fetal lung mesenchymal cells could be blocked by treating the cells with a p38 inhibitor. This suggests that p38 is the primary intracellular signaling component to mediate BMP's regulatory effects on airway myogenesis.

Despite being commonly characterized by the expression of a variety of myogenesis-associated genes, airway and vascular SMCs seem to originate from distinct mesenchymal progenitor populations (*Zhang et al., 2013*), and are regulated by distinct molecular mechanisms. Our study reveals that *Myocd* is a prominent target gene that links Bmpr1a to the genes associated with airway SMC development. However, Bmpr1a, p38, and Myocd are not critical in fetal pulmonary vascular SMC development, and inhibition of these molecules in vivo and in vitro does not have any observable effect on vascular SMCs in fetal lungs. Our previous study has demonstrated that *Myocd* is abundant in airway SMCs but barely expressed in vascular SMCs (*Young et al., 2020*), and that cardiovascular SMC development is mediated by myocardin-related transcription factors, the other members of *Myocd* family (*Parmacek, 2007*). Distinct regulation between airway and vascular SMCs is also evidenced by the phenotypes of mice with various mutations (*Badri et al., 2008*; *Murphy et al., 2008*). For instance, hypomorphic FGF-10 mice only display decreased expression of airway smooth muscle actin (*Mailleux et al., 2005*), while deletion of Wnt-7b selectively disrupts the development of vascular SMCs (*Shu et al., 2002*). Although reduced expression of Bmpr1a is associated with various forms of acquired as well as primary nonfamilial pulmonary hypertension (*Du et al., 2003*), the role of Bmpr1a-mediated signaling in regulating prenatal lung vascular SMC development has not been studied. Our current data suggest that Bmpr1a is dispensable in fetal vascular SMC development.

Previous studies have suggested that airway SMCs initially develop from local mesenchymal cells around the tips of epithelial buds undergoing bifurcation. These cells wrap around the bifurcating cleft and neck of the terminal buds, and then grow rigidly alongside the elongating bronchial tree (*Kim et al., 2015*; *Badri et al., 2008*; *Kumar et al., 2014*). Inhibition of SMC differentiation in embryonic lung explant culture, such as by blocking L-type Ca$^{2+}$ channels using nifedipine, or by inhibiting FGF and SHH signaling using SU5402SHH and cyclopamine, prevented airway terminal bifurcation (*Kim et al., 2015*). However, our current and a recently published study demonstrate that abrogation of airway SMCs by deleting mesenchymal *Myocd* does not cause any significant abnormalities in airway branching morphogenesis. This suggests that loss of SMCs is neither the primary nor sole mechanism accounting for the reduction of airway branching, dilation of terminal airways, and development of airway cysts in our *Bmpr1a* CKO lungs. Notably, the airway subepithelial elastin defects were uniquely found in the *Bmpr1a* knockout lungs with extensive cystic lesions but not in the *Myocd* CKO lungs. Interestingly, elastin expression in the developing lung parenchyma prior to alveologenesis is predominantly localized to the mesenchyme in the developing airways, as indicated by our elastin immunostaining data, suggesting that elastin plays a role in airway branching. Previous study has revealed that the perinatal development of terminal airway branches is arrested in mice lacking elastin (*Eln -/-*), resulting in dilated distal air sacs that form abnormally large cavities (*Wendel et al., 2000*). It is worthy to note that airway smooth muscles are normal in these elastin null mice, implying that loss of elastic fiber alone is not sufficient to cause airway cystic pathology, but could serve as an additional

factor contributing to the lesions in vivo. This assumption is also supported by the fact that congenital airway cysts in the lungs of human CPAM patients are characterized by compromised smooth muscle and elastin fiber (*Jiang et al., 2019*). Future studies are needed to investigate whether the combined defects of airway SMCs and elastin fibers are the pathogenic mechanisms behind congenital airway cyst formation.

The upregulation of multiple Bmp ligands, particularly Bmp4, in the *Bmpr1a* CKO lung tissues as compared with the WT lungs is a complex scenario. The isolated fetal lung mesenchymal cells of *Bmpr1a* CKO mice also showed increased Bmp4 expression as compared to the WT lung cells. This might be due to an autoregulatory feedback loop between Bmp ligand expression and intracellular BMP signaling. Previous studies have shown that disruption of BMP signaling by LDN-193189 can substantially improve Bmp4 production (*Kim and Lee, 2019*), and that deletion of Bmpr1a and Acvr1 in the lens-forming ectoderm increases Bmp2, -4, and -7 transcripts (*Huang et al., 2015*). These in vitro and in vivo studies clearly suggest the inverse relationship between mesenchymal BMP signaling activity and Bmp ligand expression. Misexpression of BMP4 results in a decrease of Sox2$^+$ proximal progenitors and an expansion of Sox9$^+$/Id2$^+$ distal progenitors in fetal mouse lungs, leading to malformed airways (*Wang et al., 2013*). Consistently, overexpression of Bmp antagonist Xnoggin or gremlin in fetal distal lung epithelium leads to a severe reduction of distal epithelial cell types and a concurrent expanded expression of the proximal epithelial cell markers CC10 and Foxj1 (*Weaver et al., 1999*; *Lu et al., 2001*). In our *Bmpr1a* CKO lungs, the proximal-distal patterning of lung airways is also substantially disturbed, as indicated by the aberrant distribution of proximal and distal epithelial cells. Evidence increasingly suggests that the lung mesenchyme is closely involved in respiratory lineage specification and epithelial differentiation by producing many critical signaling molecules, such as Bmp ligands, that direct endodermal expression of specific cell markers in a temporal and spatial fashion (*McCulley et al., 2015*; *Morrisey and Hogan, 2010*). Therefore, the decrease in proximal cell types and ectopic expression of distal cell markers (Sox9) in the *Bmpr1a* CKO lungs may be subsequently caused by abnormal increase of Bmp ligands. This aberrant epithelial growth may also contribute to airway cyst formation, as multiple mouse models have suggested that loss of Sox2$^+$ or expansion of Sox9$^+$ in airway epithelia are associated with altered epithelial differentiation and branching, as well as cystic pathology in fetal lungs (*Wang et al., 2013*; *Liberti et al., 2019*; *Rockich et al., 2013*).

In summary, mesenchymal Bmpr1a is essential for lung development, particularly airway branching morphogenesis. Disruption of Bmpr1a-mediated mesenchymal signaling causes prenatal airway malformation and cystic lesions. Multiple mechanisms, including deficiency of airway smooth muscle, airway elastin fiber defect, and perturbation of the Sox2-Sox9 epithelial progenitor axis, may contribute to the congenital airway cystic pathogenesis. In addition, the downstream Smad-independent p38 pathway appears to be critical in mediating BMP-regulated airway smooth muscle development. Our study helps to understand both normal lung development and the mechanisms of related airway malformation and congenital pulmonary diseases.

## Materials and methods
### Mice
The *Tbx4-rtTA/TetO-Cre* transgenic line was generated in our lab (*Zhang et al., 2013*). *Floxed-Bmpr1a* (*Bmpr1a* $^{fx/fx}$) mice were provided by Dr. Yuji Mishina (*Mishina et al., 2002*). *Floxed-Myocd* (*Myocd* $^{fx/fx}$) mice were provided by Dr. Michael S Parmacek (*Huang et al., 2008*; *Huang et al., 2009*). *Floxed-Smad1* (*Smad1*$^{fx/fx}$) mice were originally generated by Dr. Anita Roberts (*Huang et al., 2002*). *Floxed-Smad5* (*Smad5* $^{fx/fx}$) mice were provided by An Zwijsen at KU Leuven (*Umans et al., 2003*). *Cspg4-DsRed* mice were purchased from The Jackson Laboratory (#008241). *Tagln-YFP* reporter mice were provided by Dr. Jeffrey Whitsett at Cincinnati Children's Hospital. To generate the lung mesenchyme-specific CKO mice of genes studied in this work, *Bmpr1a*$^{fx/fx}$, *Myocd*$^{fx/fx}$, and *Smad1*$^{fx/fx}$/*Samd5*$^{fx/fx}$ mice were bred to the *Tbx4-rtTA/TetO-Cre* mice carrying *Bmpr1a*$^{fx/+}$, *Myocd*$^{fx/+}$, or *Smad1*$^{fx/+}$/*Samd5*$^{fx/+}$, respectively. Dox (625 mg/kg in food [TestDiet] and 0.5 mg/ml in drinking water [Sigma]) was given to the mothers from E6.5 to induce Cre-mediated floxed gene deletion. The littermate controls were the mice without any floxed-gene deletion due to lack of transgenes *Tbx4-rtTA* and/ or *TetO-Cre. Tagln-YFP* mice were crossed with *Cspg4-DsRed* mice to generate double fluorescence

reporter mice. Timed mating was set up as previously described (*Luo et al., 2016*). Mouse fetal lungs of different embryonic stages were collected from pregnant females. All mice were bred in C57BL/6 strain background. The mice used in this study were housed in pathogen-free facilities. All mouse experiments were conducted in accordance with NIH Guide and approved by the Institutional Animal Care and Use Committee of Children's Hospital Los Angeles (Protocol #143).

## Cell isolation, culture, and treatment

Feal lung mesenchymal cells from E15.5 lungs were isolated and cultured as previously described (*Chu et al., 2020*). To study the role of BMP signaling in subpopulations of SMCs, lung mesenchymal cells from *Tagln-YFP/Cspg4-DsRed* reporter mice were further sorted into perivascular SMCs and non-perivascular SMCs based on their fluorescent reporter expression using FACS (BD, FACSAria I). The cells were then plated at low density (~$10^4$/100 mm dish) and cultured in αMEM medium supplemented with 20% FBS, 2 mM L-glutamine, 55 μM 2-mercaptoethanol, and antibiotics (100 U/ml penicillin and 100 μg/ml streptomycin). The cells (<10 passages) were treated with recombinant mouse BMP4 protein (50 ng/ml), LDN193189 (200 nM), or p38 pathway inhibitor (SB203580, 1 μM) for 2 weeks. BMP4 protein was purchased from R&D Systems (5020 BP). LDN193189 was provided by Dr. Paul Yu at Harvard Medical School (*Yu et al., 2008*), and SB203580 was purchased from Sigma (#S8307).

## Histology and immunofluorescence analysis

Fetal mouse lungs were isolated and imaged under a dissecting microscope. Branching tips were quantified manually. For the size of branching tips, a freeform trace around each airway tip was drawn and the area within the trace was measured using ImageJ software. The data were presented as relative changes of Bmpr1a CKO to the littermate controls (*Herriges et al., 2015*). For preparation of regular paraffin tissue sections and 2D histological examination, isolated fetal lungs were fixed in 4% paraformaldehyde, as described in our previous publication (*Luo et al., 2016*). H&E and immunofluorescent staining was performed as described previously (*Luo et al., 2015*). For whole mount immunofluorescence staining and 3D imaging, fetal lungs were isolated and fixed in DMSO:methanol (1:4) overnight at 4°C, following a protocol described before (*Luo et al., 2018*). Antibodies used for immunofluorescent staining were listed in the *Supplementary file 1a*.

## Cell proliferation and apoptosis analyses

EdU incorporation was used to assess the in vivo cell proliferation following instruction from the manufacture (#C10337, Thermo Fisher Scientific). Cell quantification was performed using Fiji imaging software (1.52p) as described previously (*Luo et al., 2015*). Apoptosis was evaluated by TUNEL assay (#S7110, MilliporeSigma). E15.5 lung specimens with additional treatment of DNase I served as the positive control.

## RNA extraction and RNA-seq analysis

E15.5 mouse lungs were dissected in RNase-free PBS and flash-frozen in liquid nitrogen. After genotyping, mRNA from WT and *Bmpr1a* CKO lungs was prepared by using the TRIzol RNA extraction reagent (Thermo Fisher, #15596026). The RNeasy Micro Kit (QIAGEN, #74004) was then used to clean up the extracted mRNA. Total RNA quality was measured with a bioanalyzer (Agilent) with RIN greater than 7.0 being acceptable. RNA-seq libraries were generated and sequenced by Quickbiology (Quickbiology Inc, Pasadena, CA, USA), using the Illumina TruSeq v2 kit (Illumina, San Diego, CA, USA). Data processing was performed using the USC high-performance computing cluster (https://hpcc.usc.edu/). Roughly 50 million 100 bp single-end sequences were aligned to the Gencode M8 annotation (*Frankish et al., 2019*) based on the Genome Reference Consortium mouse genome (GRCm38.p4) and using the STAR aligner (*Dobin et al., 2013*). Read counts per gene were calculated using the HTSeq-count software (*Anders et al., 2015*). Differential gene expression was determined using the R/Bioconductor software 'edgeR' (*Robinson et al., 2010*). Genes were considered significantly different if their FDR-corrected p-values were less than 0.05 and $\log_2$FC was greater than 1.0. The sequencing data was deposited to the NCBI GEO repository with accession number GSE97946.

## Real-time PCR and western blot

Lung mesenchymal cells and tissue specimens were harvested and flash-frozen in liquid nitrogen for the subsequent mRNA and protein analysis. Total RNA was isolated from the cultured cells and

lung tissues using the RNeasy Kit (QIAGEN, #74106). The cDNA synthesis and real-time PCR were performed as described in a previous publication (*Luo et al., 2015*). The related oligonucleotide primers are listed in *Supplementary file 1b*. Protein lysate was prepared from the cells and lung tissues (n>3 per group) and analyzed by western blot as previously described (*Chu et al., 2020*). The related antibodies used for western blot are listed in *Supplementary file 1a*.

## Statistical analysis

The quantitative data are presented as means ± SD. All in vitro experiments were repeated at least three times, and data represent consistent results. At least five mice per experimental group were utilized, which have 80% power to detect a 20% difference in lung phenotypes with a 0.05 two-sided significance level. The statistical difference between any two groups was assessed by an independent sample t-test. One-way analysis of variance (ANOVA) was used to examine differences between groups of three or more, followed by post hoc comparisons to study differences among individual group. Chi-square test with Yates correction was used for GO enrichment analysis. $p < 0.05$ was considered statistically significant.

## Acknowledgements

We thank Dr. Robert Mecham at Washington University, St. Louis, Dr. Paul Yu at Harvard University, Dr. Jeffrey Whitsett at Cincinnati Children's Hospital, and Dr. An Zwijsen at KU Leuven for providing the mouse lines and other reagents. Dr. Esteban Fernandez at the Cell Imaging Core of Children's Hospital Los Angeles helped with the confocal imaging. This study was supported by NIH grants HL151699 and HL141352 (WS).

## Additional information

### Funding

| Funder | Grant reference number | Author |
|---|---|---|
| National Heart, Lung, and Blood Institute | HL151699 | Wei Shi |
| National Heart, Lung, and Blood Institute | HL141352 | Wei Shi |

The funders had no role in study design, data collection and interpretation, or the decision to submit the work for publication.

### Author contributions

Yongfeng Luo, Conceptualization, Data curation, Formal analysis, Supervision, Validation, Investigation, Visualization, Methodology, Writing – original draft, Project administration, Writing – review and editing; Ke Cao, Data curation, Formal analysis, Validation, Investigation, Methodology; Joanne Chiu, Hui Chen, Investigation, Methodology; Hong-Jun Wang, Investigation; Matthew E Thornton, Formal analysis, Investigation, Visualization, Methodology; Brendan H Grubbs, Resources, Formal analysis; Martin Kolb, Formal analysis, Writing – review and editing; Michael S Parmacek, Resources, Writing – review and editing; Yuji Mishina, Conceptualization, Resources, Writing – review and editing; Wei Shi, Conceptualization, Resources, Formal analysis, Supervision, Funding acquisition, Validation, Investigation, Visualization, Methodology, Writing – original draft, Project administration, Writing – review and editing

### Author ORCIDs

Yongfeng Luo http://orcid.org/0000-0001-8765-0273
Matthew E Thornton http://orcid.org/0000-0002-1083-2703
Michael S Parmacek http://orcid.org/0000-0003-1449-4665
Yuji Mishina http://orcid.org/0000-0002-6268-4204
Wei Shi https://orcid.org/0000-0001-6499-2473

## Ethics

All mouse experiments were conducted in accordance with NIH Guide and approved by the Institutional Animal Care and Use Committee of Children's Hospital Los Angeles (Protocol #143).

Reviewer #2 (Public Review): https://doi.org/10.7554/eLife.91876.3.sa1
Author response https://doi.org/10.7554/eLife.91876.3.sa2

## Additional files

### Supplementary files
- MDAR checklist
- Supplementary file 1. Antibody and Primer Resources.

### Data availability

The sequencing data have been deposited to the NCBI GEO repository with accession number GSE97946. All data generated and analyzed during this study are included in the manuscript and supporting files.

The following dataset was generated:

| Author(s) | Year | Dataset title | Dataset URL | Database and Identifier |
|---|---|---|---|---|
| Shi W, Luo Y, Thornton ME | 2024 | Next-generation RNA Sequencing and Transcriptome Comparison of Mouse Wild Type and Mesenchymal Bmpr1a conditional knockout lungs of E15.5 | https://www.ncbi.nlm.nih.gov/geo/query/acc.cgi?acc=GSE97946 | NCBI Gene Expression Omnibus, GSE97946 |

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
