## [Editor Report · eLife assessment]

This **valuable** paper characterizes a murine model for congenital cystic airway abnormalities (CPAM). In contrast to previous assumptions that only epithelial cells are involved in the formation of pulmonary cysts, the authors provide **compelling** new evidence that defective BMP signaling in lung mesenchymal cells can disrupt airway development. Knowing that proper BMP signaling in mesenchymal cells is required for normal cyst-free lungs could potentially pave the way to understanding and preventing CPAM in infants at risk for this common disorder, which can be fatal if untreated. The relevance of the murine model could be enhanced by further molecular and histological comparison with human cysts.

---

## [Referee Report · Reviewer #2 (Public Review)]

Congenital cystic airway abnormalities (CPAM) are a common poorly understood disorder in airway lung development that can be fatal if not effectively treated at birth. This study by Luo and colleagues provides compelling new evidence that bone morphogenetic protein signaling in distal mesenchymal cells is required for normal mouse lung development. Genetic loss of BMP receptor in mice and in fetal mesenchymal cells causes type 2 or alveolar-like CPAM pathology. Furthermore, this is associated with changes in expression of Sox2-Sox9 suggesting defects in the proximal to distal cellularity of the lung. Interestingly, cysts are formed even when SMAD1 and 5, two major downstream effects of BMP signaling are deleted suggesting a role for non-canonical BMP signalling. Furthermore, they were independent of ablating BMP signaling in non-vascular mesenchymal cells. The findings are compelling and provide strong evidence that cystic lung development is caused by loss of non-canonical BMP signaling in mesenchymal cells. The main weakness of the paper is that it does not identify the downstream non-canonical effector of mesenchymal BMP signaling. The authors provide a plausible suggestion that it may be p38 MAPK that deserves further investigation. Despite this minor weakness, the overall findings are novel and considered important because they provide a foundation for new studies, including experiments that may produce drugs designed to prevent or treat newborn infants with CPAM.

---

## [Author Response]

The following is the authors’ response to the original reviews.

**Reviewer #1:**
(1) Importantly, it would be useful to have provided more detailed information on the structure and histological properties of the murine cysts and how such findings relate to human lung cysts. Also, the authors should examine whether there is any information on Bmpr1a in human cyst formation (i.e GWAS data).

We fully agree that it is important to examine Bmpr1a in human cyst pathology. Unfortunately, there is no GWAS data on this. From the published RNA-seq data, which were obtained from postnatal lung specimen of congenital pulmonary airway malformation (CPAM) patients, “integrated suppression of BMP signaling pathway” was reported although altered expression of BMPR1A was not presented. We speculate that (1) BMPR1A is critical in embryonic development and a germline deficiency of BMPR1A may lead to early embryonic lethality prior to lung formation as supported by mouse data; (2) As suggested by our previously published study related to TGF-beta signaling and prenatal pulmonary cysts (Miao et al., Am J Physiol Lung Cell Mol Physiol 2021), dysregulation of BMPR1A-mediated signaling in a particular time window of fetal lung development may be sufficient to cause cyst formation, so that BMPR1A alteration may not be persistent to postnatal lung specimens.

(2) Throughout the paper, there is a lack of quantification for the histological findings. Littermate controls should also be clearly defined genetically,

We thank the reviewer for this suggestion and acknowledge the importance of quantitative measurement for the changes. We now add quantitative data on branching number and size of the airway tips to define the difference between wild-type and Bmpr1a CKO mouse lungs in Fig.1.“The littermate controls were the mice without any gene deletion due to lack of transgenes Tbx4-rtTA and/or TetO-Cre”, which is now added in Materials and Methods.

(3) Figure 1 suppl: "Doxycycline" is misspelled.

This has been corrected.

(4) Figure1c Suppl: Hard to discern clear-cut expression of Bmpr1a protein in mesenchyme in WT. Comparable images with similar sizes of airways should be used.

To provide a clearer comparison of Bmpr1a expression patterns between Bmpr1a CKO and control lungs, we enlarge the fluorescent stained lungs presented in Supplemental Figure 1C as suggested by the editor. Additionally, dotted lines have been added to delineate the airway boundaries from the surrounding mesenchyme to better visualize the Bmpr1a distribution in lung mesenchyme. Bmpr1a expression in fetal lung mesenchyme is easily detected at E15.5 when significant dilation of airways is presented in Bmpr1a CKO lung. It is rare to have comparable sizes of peripheral airways in the Bmpr1a CKO lung at this point.

(5) Figure 2a: Expression of several genes studied and altered should be identified on scatter plot.

As suggested by the reviewer, we now highlight the related genes, including Acta2, Myocd, Eln, Bmp4, Sox2, etc., in the scatter plot. In addition, we also highlight these critical genes in the heatmap (Fig. 2B and Fig. 7B).

(6) Figure 2c: Authors should also consider staining for other smooth muscle markers.

We now include a panel of Myh11 immunostaining in Figure 2E. Myh11 is another common marker for smooth muscle cells. Lack of Myh11 staining in Bmpr1a CKO lung airways further supports our conclusion that loss of mesenchymal Bmpr1a leads to defective airway smooth muscle growth.

(7) Figure 3: ELN expression should be defined in a clear quantitative manner.

We have presented RNA-seq data, Real-time PCR results, immunostaining, and western blot data for in vivo samples. Additionally, we have included in vitro experiment illustrating that Bmp4 induces Eln expression, suggesting that BMP signaling regulates Eln expression. We believe that these datasets collectively support our conclusion.

(8) Figure 4: Additional information on p38 dependent signaling (Including in vivo studies) would potentially help to understand key molecular events and perhaps could help to address key mechanistic events, including their location and identity.

We sincerely appreciate the insightful suggestion from the reviewer. While the study of p38-dependent signaling is definitely important to dissect the entire mechanisms, we are not going to include such experiments in this manuscript due to time constraints associated with in vivo studies.

(9) Figure 6: Would be helpful to know whether Bmpr1a receptor is expressed in Myocd KO.

Bmpr1a expression is not changed in Myocd KO lungs, which is now included as Figure 6C. Together with other data, this suggests that Myocd is a downstream target directly mediating Bmpr1a-regulated airway smooth muscle development.

(10) Figure 7: Not clear how these findings, though interesting, relate to the body of studies and the pathogenesis of cyst formation. Other points: (1) The authors should re-examine/repeat co-staining in the KO mouse lung (right 2 images in the top group of 4) for Foxj1, Sox2, and CDH (right 2 images, Figure 7A). For one thing, the cadherin stain in the 2 KO images seems localized to the lumen. Secondly, the pattern of cadherin staining looks exactly the same in both KO images, suggesting an error and/or duplication (2) authors should place arrows on the heat map showing the location of SPC, Sox2, Sox9, and FoxJ1 bands (3) figure 7D graph needs numbers on y axis.

Fig.7 provides an additional potential mechanism by which deficient Bmp signaling leads to abnormally increased Bmp ligand expression, which disrupts the formation of epithelial proximal-distal axis, and results in cystic defects. Further in vivo experiments are needed to test this, which is beyond the scope of this paper.

The E-cadherin staining signal in the lumen is caused by the tissue section positioned at an interface between lumen and the apical membrane of the lining epithelial cells where the E-cadherin is localized.

Triple immunostaining of E-Cadherin, Sox2, and FoxJ1 was performed for the same tissue section (upper two panels of Figure 7A) as these antibodies were derived from different species, but the images are presented in two different combinations for simplicity and clarity. For the lower two panels of Figure 7A, double immunostaining of Sox9/E-Cadherin and Spc/E-Cadherin were performed separately on different tissue sections due to both anti-Sox9 and anti-Spc antibodies were produced from rabbits.

The genes listed in the heatmap are canonical and putative marker genes for differential lung epithelial cell lineages, such as Scgb1a1 for Clara cells and FoxJ1 for ciliated cells. Therefore, progenitor cell marker Sox2 and Sox9 were not included. In the updated heatmap, four widely acknowledged epithelial cell markers—Scgb1a1, FoxJ1, Sftpb, and Sftpc have been distinguished by utilizing a distinct font color (red) to enhance their visibility.

Label for the y axis of Fig.7D is now added.

**Reviewer #2 (Public Review):**
(1) The authors may be aware that a recent paper (https://doi.org/10.1038/s41598-022-24858-3) reported on transcriptional changes seen in human CPAM. It would seem that some of the molecular changes seen in human CPAM move in the opposite direction of what is reported in mice lacking mesenchymal Bmrp1a. Perhaps the authors could comment on these differences in the discussion and whether they potentially explain the etiology of CPAM or branching morphogenesis in general.

We thank the reviewer for referring this paper regarding human CPAM study. CPAM has a variety of histopathology. The type 1 CPAM is assumed to develop from more proximal bronchial/bronchiolar airways while type 2 CPAM is developed from relatively distal bronchiolar airways. In that publication, surgical resected lung specimens were collected from type 1 CPAM patients postnatally (0.5-1 year), in which the cysts were lined with ciliated pseudostratified columnar epithelial cells. Gene expression was compared between cystic lung tissues and adjacent non-cystic lung tissues. Interestingly, integrated suppression of BMP signaling pathway was shown by their data analysis. In our mouse model, the histopathology appears as human type 2 CPAM, such as back-to-back cysts lining with a simple layer of epithelial cells. Therefore, several factors could explain the differences between their published data and our study at the molecular level: (1) Different types of CPAM based on the histopathology; (2) Different sampling time points, developing cysts at fetal stage in mouse sample vs. developed cysts in postnatal huma samples; (3) Different comparison of diseased and normal tissues: separate normal lungs vs. cystic lungs in mice while in human cystic tissues vs. non-cystic tissues in the same lungs. We now include this reference in the Discussion.

(2) Figure 4 shows that BMP4 increases SMADs, p38, and several muscle genes in mesenchymal cells. Figure 5 extends this finding with a clever strategy to label airway and vascular smooth muscle with different fluorescent molecules used to isolate different types of mesenchymal cells. It shows that non-vascular smooth muscle cells but not perivascular smooth muscles are responsive to BMP4 signaling as defined by increased expression of Myh11. Are there cell-restricted responses to the other genes shown in Figure 4? Given the lack of SMAD signaling and the increase seen in p38 signaling, would blocking p38 signaling influence the BMP responsiveness of these nonvascular smooth muscle cells?

We thank the reviewer for this constructive comment. As we have addressed above, we will leave p38-mediated signaling and cyst formation to next step study due to time constraints associated with these studies.

(3) Figure 6 shows that mesenchymal loss of Myocd causes a deficiency of airway smooth muscle cells, but this was not sufficient to create cysts. Did the authors ever check to see if it changed Sox2-Sox9 staining in the airway epithelium?

There is no significant change in Sox2 expression in proximal airway epithelia of Myocd CKO lungs as detected by immunostaining. The result was not included in this manuscript.

(4) Figure 7 shows that mesenchymal loss of Bmpr1a proximalizes the distal airway as defined by loss of Sox2 and FoxJ1 (a ciliated marker) and gain in (Sox9 and SP-C) staining. But Club cells expressing Scgb1a1 and Cyp2F2 are the predominant epithelial cells in the distal airway. The transcriptomics data in panel B shows expression of these genes is less in the mutant mice. Does this mean they fail to generate Club cells or there is just less expression per cell? In other words, what are the primary epithelial cells present in the airways of mice with loss of mesenchymal Bmpr1a?

As shown in the heatmap of Fig.7b, the dysregulated gene expression in the Bmpr1a CKO extends beyond the featured epithelial cell markers, encompassing alterations in numerous putative marker genes. For example, several putative Club cell markers in addition to Scgb1a1 and Cyp2F2 were reduced in the Bmpr1a CKO lungs, suggesting a compromised differentiation of Club cells. Additionally, we observed upregulations of some molecular markers for distal progenitors and differentiated cells in the proximal region of airways, again suggesting a significant disruption in epithelial differentiation in the Bmpr1a CKO lungs. These abnormal cells can be further defined by a single cell transcriptomic approach in future.

**Recommendations for Authors:**

**Reviewer #1 (Recommendations For The Authors):**
As discussed above, there may be an issue with the histological images and staining in 2 images in Figure 7A. The precise images, problems and suggestions to resolve the issue are in the Review.

Please see our response to Reviewer 1 above.

**Reviewer #2 (Recommendations For The Authors):**
Minor Weaknesses:(1) Please enlarge the fluorescent stained lungs presented in Supplemental Figure 1C.

We have revised this panel accordingly.

(2) Figure 1D and E show that loss of Bmpr1a does not change proliferation or apoptosis on E15.5. Was that also seen through E18.5?

We thank the reviewer for the thoughtful question about proliferation and apoptosis at later embryonic stages. Our focus here was to elucidate the mechanisms underlying abnormal branching morphogenesis and lung cyst initiation that occur prior to E15.5 in our model. Measuring the dynamic changes in cell proliferation and apoptosis at later timepoints will help to understand cyst progression, which will be our next focus.

(3) BMP inhibitors used in Figure 4 show that BMP signaling regulates mesenchymal myogenesis independent of SMAD. But the experiments don't show how the inhibitors impact the control cells.

We have examined the effects of the BMPR1 inhibitor LDN on the control cells. At the same dose (200 nM) and serum-free culture condition, LDN did not affect the basal level of BMP signaling (data not included) but blocked exogeneous BMP4-induced signaling elevation (Fig.4E).

(4) Bmpr1a was deleted by administering doxycycline to pregnant dams prior to lung bud formation. It caused cystic disorders by disrupting proximal airspace. Could the authors speculate on why it does not impact tracheal and bronchiolar development? In other words, does the TBX4 promoter not target these cells? Do these cells not express Bmpr1a?

The Tbx4 enhancer does target mesenchymal cells surrounding the trachea and bronchioles. Deletion of Bmpr1a in tracheal mesenchymal cells result in disruption of tracheal cartilage formation and smooth muscle differentiation. These phenotypes are evident in the gross view of lungs from E15.5 and later (Fig.1A). However, our manuscript is focusing on the phenotype of prenatal lung cysts, and we have chosen not to include complex data on tracheal development.